

# Pangenomic type III effector database of the plant pathogenic *Ralstonia* spp.

Cyrus Raja Rubenstein Sabbagh[1,*], Sebastien Carrere[1,*],
Fabien Lonjon[2], Fabienne Vailleau[1], Alberto P. Macho[3],
Stephane Genin[1] and Nemo Peeters[1]

[1] LIPM, Université de Toulouse, INRA, CNRS, Castanet-tolosan, France
[2] Department of Cell & Systems Biology, University of Toronto, Toronto, ON, Canada
[3] Shanghai Center for Plant Stress Biology, CAS Center for Excellence in Molecular Plant Sciences, Shanghai Institutes of Biological Sciences, Chinese Academy of Sciences, Shanghai, China
* These authors contributed equally to this work.

Corresponding author
Nemo Peeters, nemo.peeters@inra.fr

## ABSTRACT

**Background:** The bacterial plant pathogenic *Ralstonia* species belong to the beta-proteobacteria class and are soil-borne pathogens causing vascular bacterial wilt disease, affecting a wide range of plant hosts. These bacteria form a heterogeneous group considered as a "species complex" gathering three newly defined species. Like many other Gram negative plant pathogens, *Ralstonia* pathogenicity relies on a type III secretion system, enabling bacteria to secrete/inject a large repertoire of type III effectors into their plant host cells. Type III-secreted effectors (T3Es) are thought to participate in generating a favorable environment for the pathogen (countering plant immunity and modifying the host metabolism and physiology).
**Methods:** Expert genome annotation, followed by specific type III-dependent secretion, allowed us to improve our Hidden-Markov-Model and Blast profiles for the prediction of type III effectors.
**Results:** We curated the T3E repertoires of 12 plant pathogenic *Ralstonia* strains, representing a total of 12 strains spread over the different groups of the species complex. This generated a pangenome repertoire of 102 T3E genes and 16 hypothetical T3E genes. Using this database, we scanned for the presence of T3Es in the 155 available genomes representing 140 distinct plant pathogenic *Ralstonia* strains isolated from different host plants in different areas of the globe. All this information is presented in a searchable database. A presence/absence analysis, modulated by a strain sequence/gene annotation quality score, enabled us to redefine core and accessory T3E repertoires.

## INTRODUCTION

Plant pathogenic *Ralstonia* species (*Peeters et al., 2013b*) were ranked among the 10 most important plant bacterial pathogens (*Mansfield et al., 2012*). These soil-resident bacteria are indeed important, as they affect many different plant species, ranging from solanaceous crops to other important crops like banana and peanut, in different parts of the world. Recently, new plant species have been found to be infected and present

symptoms of bacterial wilt, like blueberry shrubs in Florida, USA (*Bocsanczy, Espindola & Norman, 2019*), ornamental roses in the Netherlands (*Bergsma-Vlami et al., 2018*), or pumpkin in China (*She et al., 2017*). This bacterium has one of the largest known repertoires of T3Es among all plant or animal pathogenic bacteria. The type III secretion system (T3SS) of Gram negative phytopathogenic bacteria is essential for virulence, and Type III-secreted effectors (T3Es hereafter) have been found to contribute in many different and sometimes redundant manners to the fitness of the bacterium in interaction with its host (*Buttner, 2016*).

Plant pathogenic *Xanthomonas* spp., and animal pathogens like *Escherichia* spp., *Shigella* spp. or *Yersinia* spp. have around 30 T3Es per strain (*Dong, Lu & Zhang, 2015*; *Schwartz et al., 2015*). Classically known strains of *Pseudomonas* spp. have also around 30–40 T3Es (*Wei et al., 2015*), with some rare cases of up to 50 T3Es in a given strain (*Dillon et al., 2019*). It was reported that *Legionella* spp. can secret in their host cells up to 300 effectors type IV effectors (*Gomez-Valero et al., 2019*). Plant pathogenic *Ralstonia* spp. have between 46 and 71 T3Es (*Peeters et al., 2013a*).

In this work, we curated the genome of two new phylotype I strains bringing the total number of curated strains to 12 plant pathogenic *Ralstonia* strains, representing the known diversity of phylotypes (*Wicker et al., 2012*), more recently subdivided into three species (*Safni et al., 2014*). This generated new and updated profiles for the prediction of 102 Rips ("*Ralstonia* injected Proteins") and 16 hypothetical Rips, to be compared with the previous 94 Rips and 16 hypothetical Rips (*Peeters et al., 2013a*). Two hypothetical Rips from the reference strain CMR15, Psi07, and GMI1000 were experimentally confirmed as being *bona fide* Rips (and were named RipBM and RipBO).

The new and improved prediction profiles were used to analyze the effector repertoires of the 155 genomic sequences available in genbank. This dataset represents 140 different strains spread over the three newly defined species: 54 *Ralstonia solanacearum* stains (16 Phylotype IIA and 38 Phylotype IIB strains), 59 *R. pseudosolanacearum* strains (57 Phylotype I and two Phylotype III strains) and 27 *R. syzygii* strains (27 Phylotype IV strains). The prediction of all 118 Rips (including hypothetical Rips) over the whole dataset of 155 genomes/140 different strains is available as a browsable database, enabling direct comparisons between strain repertoires, from presence/absence tables to multiple alignments of DNA and protein sequences. This dataset was then further analyzed to evaluate how conserved the Rips are among these 140 strains. This analysis took into account the host of isolation as a strong (but limited) host cue, or the phylogenetic group, to identify host or kinship repertoire conservation.

## MATERIALS AND METHODS

### 155 published whole genomes

The genbank genome data repository was scanned for the presence of complete genome sequences of *Ralstonia* species complex strains. The total number of genomes gathered was 155, with some strains sequenced multiple times by different research groups, yielding sequence data for 140 distinctive strains. Owing to the fact that for a same strain different isolates could be slightly different, and also to the fact that sequence quality is

important for gene repertoire completeness, we decided to keep all strain duplicates (in the database duplicates and triplicates are indicated as "−2" and "−3," respectively). Strains in duplicates are the following: FJAT-1458, FJAT-91, PSS4, CFBP2957, K60, CFBP6783, IBSBF1503, IPO1609, Po82, and UW163. Molk2 strain was present in the database with three independent sequence files, and UW551 with four independent sequence files. Table S1 contains all the available data on the 155 genome files. Whenever available, data for the following fields were also recorded: strain synonym; pubmed ID of reference articles; species name (*Safni et al., 2014*); phylotype; geographical origin (isolation site); plant isolated from; genome assembly size; assembly score; number of contigs; number of scaffolds; bioproject. Fig. S1 provides a *mutS* phylogeny (*Wicker et al., 2012*) indicating the strain relatedness.

## Genome quality

Some genome sequences deposited by their authors were of insufficient quality to be included in the Refseq database. Among the different quality criteria used by refseq one can find low contig N50, low gene count, or absence of essential rRNA, robosomal proteins and tRNA genes; for a complete list of criteria, see https://www.ncbi.nlm.nih.gov/assembly/help/anomnotrefseq/. This is the case for the genomic entries FJAT-452, FJAT-462, T110, T12, T25 and UW700. These sequences were left on the complete database but were excluded for the further analysis in this work.

We then devised an assembly score in order to sort all the strains and to distinguish draft from complete and "polished" genomes. This score is $Log_{10}$ (assembly size/number of contigs, with contigs being the N-free scaffolds or assembled pseudomolecules that were spliced by us on N stretches), and is a good general score to assess the overall completeness of the genome sequence. One exception to this is the strain CFBP2957, with a high score (6.287), but for which only the chromosome was available (and not the megaplasmid, see the bipartite nature of plant pathogenic *Ralstonia* genomes (*Salanoubat et al., 2002*)), and thus artificially increased the quality score.

We then used a known metric for genome quality called Benchmarking Universal Single-Copy Orthologs ("BUSCO") based on the comparison of predicted ORF (we used PRODIGAL; *Hyatt et al., 2010*) with an adequate set of conserved single-gene orthologs (*Waterhouse et al., 2017*). We used the beta-proteobacteria lineage derived set of 582 BUSCO genes. The completeness metric (C%) represents the presence and not-frameshifted BUSCO ORFs in a given strain genome. This data was added in Table S1. This metric was not sufficient to remove the poorly sequenced genomes as five out of the six genomes excluded from the Refseq database had a high completeness score (T12: 98.8%; T110: 98.6%; UW700: 98.6%; FJAT-462: 98.6%; T25: 96.2%).

Neither the assembly score, the BUSCO metric (nor a combination of both) was efficient enough to weed out poorly sequenced genomes. We thus felt the need to indirectly rate the gene annotation and prediction if it were to be further used in T3E repertoire comparisons. This is why we decided to generate two stringency cutoffs using the total T3E gene prediction performed by our prediction pipeline: for each strain the content in multicopy paralogous genes ("MULTI"), the single defined genes ("OK"), the

frameshifted genes ("FS"), and the pseudogenes ("PG"), were computed for the 102 T3Es. We applied two levels of stringency. For "stringency 1," we kept only the strains for which the total number of pseudogenes plus frameshifted genes is lower than 10: (PG + FS) < 10; this yielded a total of 123 genomes corresponding to 114 different strains. For "stringency 2," we only kept the strains that also had more than 50 T3Es in total; this yielded to a set of 88 genomes corresponding to 84 different strains. Table S1 contains two columns identifying the 123 "stringency 1" and 88 "stringency 2" strains.

This "stringency" ranking is an artificial cutoff, but we believe this is a valid method to further compare the complete gene repertoires. The two strains T110 and UW700 have high genome assembly scores (6.45 and 5.06 respectively), but performed badly in this stringency test, with only 30 and 21 well-predicted T3Es (excluding them from "stringency 2" group) and 35 and 25 frameshifted and pseudogenes (excluding them from "stringency 1" group).

## Gene presence/absence

For each strain, the Table S1 contains the presence/absence scoring for all the 102 Rips and 16 hypothetical Rips. We used the prediction data for each strain (see further) as highlighted on the database website (www.ralsto-T3E.org). Frameshifted genes are rare in well-sequenced genomes. Indeed, out of the 67 strains reported with two scaffolds (corresponding to the expected chromosome and megaplasmid; *Salanoubat et al., 2002*), 52 have no frameshifted genes, and nine only contain one frameshifted gene (see Table S1 for the data). We thus hypothesized that a frameshift is more due to sequencing errors than representing true genomic data. As a consequence, when making a binary scale for scoring the presence/absence of T3Es, we considered all "MULTI" (recently duplicated genes), "OK" (single gene) and "FS" (frameshifted) as "1" (or "present"); when absence "NO" and pseudogene ("PG") were considered as "0" (or "absent"), this same reasoning was used previously (*Peeters et al., 2013a*).

## T3E prediction improvements

We have improved our first T3E prediction pipeline (*Peeters et al., 2013a*), by adding databases of confirmed T3Es from *Xanthomonas* spp. (www.xanthomonas.org/t3e.html) and *Pseudomonas* spp. strains (www.pseudomonas-syringae.org/T3SS-Hops.xls). In order to capture more distantly related Rips, we lowered the tblastN/blastX thresholds (query coverage per subject 60% and percentage of identical matches 60%), this was well exemplified by the RipBN case, an AvrRpt2 ortholog clearly present in the CMR15strain (*Eschen-Lippold et al., 2016*) and also detectable by blast, but not without slightly lower thresholds. We also rewrote some parts of the pipeline in order to speed up the prediction engine. The updated pipeline is outlined in Fig. 1.

## T3Edb v3 specificities

The new database version is very similar to the previous version (*Peeters et al., 2013a*). In this new version a set of curated strains ("curated repertoire") is listed on a tab, with a comparison of their T3E repertoire. This set is composed of the following strains: 244 (phylotype I)

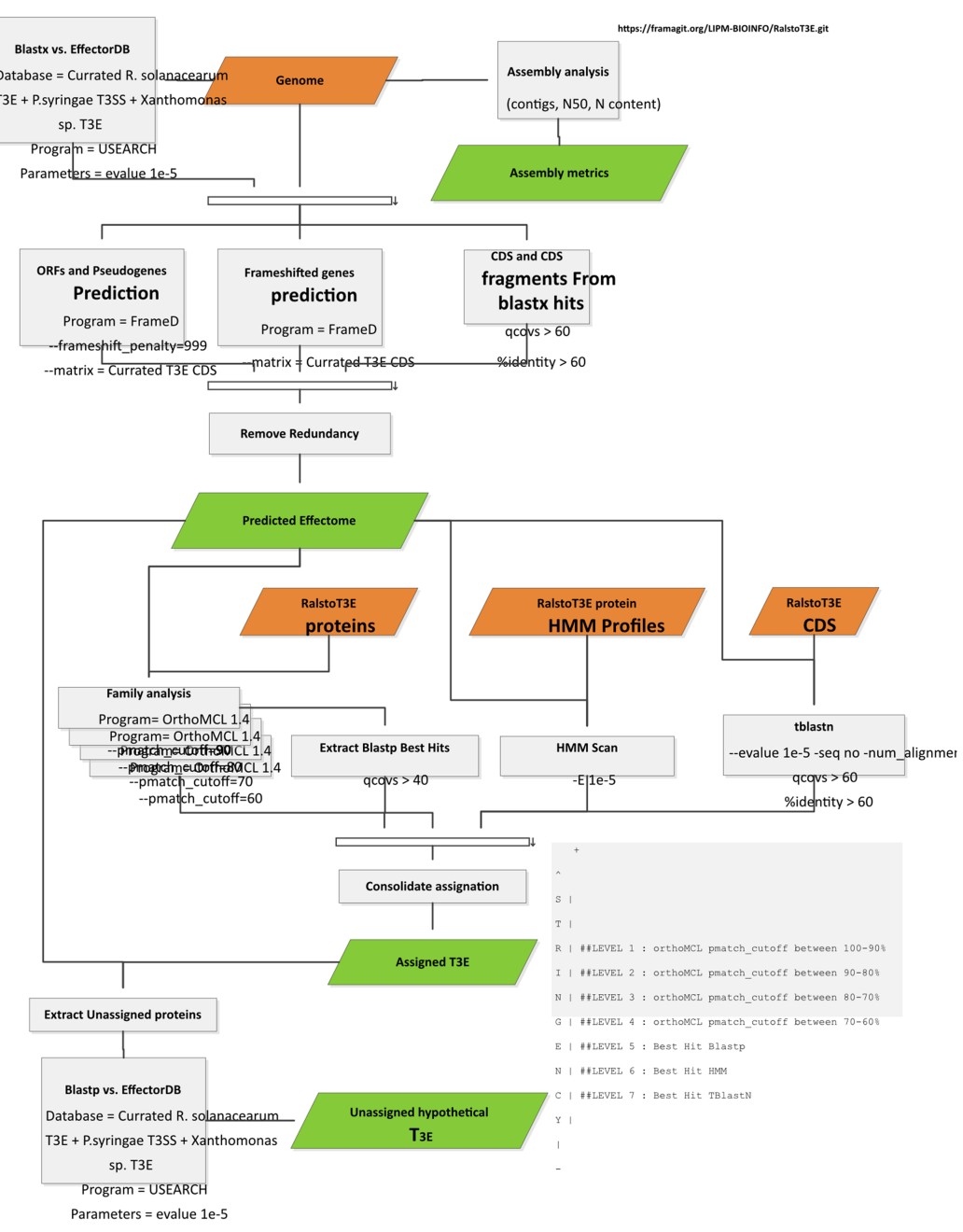

**Figure 1 T3E prediction pipeline.**

(*Ramesh et al., 2014*); GMI1000 (I) (*Salanoubat et al., 2002*); YC45 (I) (*She et al., 2015*); CFBP2957 (IIA) (*Remenant et al., 2010*); CMR15 (III) (*Remenant et al., 2010*); IPO1609 (IIB) (*Gonzalez et al., 2011*); Molk2 (IIB) (*Remenant et al., 2010*); Po82 (IIB) (*Xu et al., 2011*); UW551 (IIB) (*Gonzalez et al., 2011*); PSI07 (IV) (*Remenant et al., 2010*); BDBR229 (IV) (*Remenant et al., 2011*); and R24 (IV) (*Remenant et al., 2011*).
In order to "build profiles" of Rip prediction in different strains to compare the strains and/or to generate multifasta files (of nucleotide or protein sequences of specific Rips), one can now sort the whole set of 155 complete genomes on the different headers available, namely these are: "status" (curated or not); "code" (abbreviated name); "synonym", "species name" (*Safni et al., 2014*); "phylotype"; "plant isolated from"; "assembly size"; "number of contigs"; "number of scaffolds"; "assembly score" (see definitions above).

## Type III secretion dependence

The type III-dependent secretion of Hyp15 and Hyp16, two hypothetical T3Es previously identified in strains GMI1000, CMR15, and PSI07 (*Peeters et al., 2013a*) was demonstrated in this work. The coding sequences of PSI07_1860 and CMR15v4_mp10184 (both formerly Hyp15), were ordered as DNA synthesis from Sangon (Shanghai, China). RSc3174, from the reference strain GMI1000 (formerly Hyp16) was amplified in two steps. The first PCR was performed using the following primers: Forward: 5′GGAGATAG AACCATGAAAGTCGGCAACCAATC-3′ and Reverse 5′CAAGAAAGCTGGG TCTCCACGTGATAAGTTGTAGCG-3′, using proof-reading Phusion polymerase using high GC buffer (New England Biolabs, Ipswich, MA, USA). The second PCR was performed using one μl of the first PCR as matrix and attB universal primers (oNP291 5′ GGGGACAAGTTTGTACAAAAAAGCAGGCTTCGAAGGAGATAGAACCATG-3′ and oNP292 5′-GGGGACCACTTTGTACAAGAAAGCTGGGTC-3′, using the same polymerase as the previous PCR with a two step annealing temperature: 10 cycles at 45 °C and then 25 cycles at 55 °C. Then, Rsc3174, PSI07_1860 and CMR15v4_mp10184 were cloned into pDONR207 vector using a BP reaction and in pNP329 using a LR reaction following the instructions of the manufacturer (LifeTechnologies, Carlsbad, CA, USA). The final expression vectors were transformed into the *R. pseudosolanacearum* GMI1000 strain and in the *hrcV* mutant (type III secretion defective mutant, used as a negative control) as previously described (*Perrier, Barberis & Genin, 2018*). In-vitro secretion assays and western blot analysis were performed as previously described (*Lonjon et al., 2018*).

## RESULTS

### Curation of two new phylotype I strains and identification of eight new Rips

Because strain GMI1000 was the only curated *R. pseudosolanacearum* strain in the former RalstoT3Edb (*Peeters et al., 2013a*), we conducted a manual curation of the Type III effectome in two other *R. pseudosolanacearum* strains, both differing in host range from GMI1000. Strain Rs-10-244 was isolated from chilli pepper (*Capsicum annuum*) on the Andaman Islands (India) (*Ramesh et al., 2014*) and strain YC45 was isolated from a monocotyledoneous host, aromatic ginger (*Rhizoma kaempferiae*) in Southern China (*She et al., 2015*). Manual curation identified 73 Rip genes (+1 candidate) in strain YC45 and 77 Rip genes (+3 candidates) in Rs-10-244. Novel Rip effectors and candidates were identified in these strains (Table 1).

RipBJ was identified by secretome analysis of the GMI1000 strain (*R. pseudosolanacearum*) (*Lonjon et al., 2016*). RipBK and RipBL were identified in the process of curation of

**Table 1 Eight new T3E and two new hypothetical T3E identified.**

| Proposed T3E family name | Representative gene member | Hop/Xop homologues | Functional domain | Evidence for T3SS-dependent secretion or translocation |
|---|---|---|---|---|
| RipBJ | GMI1000 RSp0213 | none | | *Lonjon et al. (2016)* |
| RipBK | YC45_c025370 | HopAM1 | | *Chang et al. (2005)* |
| RipBL | YC45_m001910 | HopAO1 | Protein-tyrosine phosphatase | *Chang et al. (2005)* |
| RipBM | Psi07 RSPsi07_1860 (former Hyp15) | | Protein-Ser/Thr kinase | This work |
| RipBN | CMR15v4_30917 | AvrRpt2 | Cysteine protease | *Eschen-Lippold et al. (2016)* |
| RipBO | GMI1000 RSc3174 (former Hyp16) | none | | This work |
| RipBP | OE1-1_24290 | HopW1 + homologs in Xanthomonas | N-term domain = HopW1 and C-term = uncharacterized protein ABJ99_3552 (*Pseudomonas syringae* pv. cilantro) | *Zumaquero et al. (2010)* |
| RipBQ | KACC10722_38580 | HopK1/XopAK | | *Chang et al. (2005)* |
| Hyp17 | RS244_m000380 | none | | This work |
| Hyp18 | CMR15v4_mp10535 | none | | This work |

the strain YC45 (*R. pseudosolanacearum*), owing to their similarity to HopAM1 (*Chang et al., 2005*; *Goel et al., 2008*) and HopAO1 (*Chang et al., 2005*; *Macho et al., 2014*). RipBM and RipBO, formerly known as Hyp15 and Hyp16, respectively (*Peeters et al., 2013a*), were experimentally confirmed to be secreted by the T3SS in GMI1000 (Fig. 2; Fig. S2). RipBN was identified by sequence homology in the strain CMR15 (*R. pseudosolanacearum*) (*Eschen-Lippold et al., 2016*). RipBP (homolog to HopW1 (*Zumaquero et al., 2010*)) was identified in the strain OE1-1 (*R. pseudosolanacearum*) and RipBQ (homolog to HopK1; *Chang et al., 2005*) in the strain KACC10722 (*R. syzygii*). RipBP and RipBQ are considered here as Rips by applying the rule of similarity with a known T3E (*Peeters et al., 2013a*). These two latter Rips have been highlighted in the curated list of strains although none of these curated strains harbor these effectors. This is the same for RipBE which is specific to strain RS1000 (*Mukaihara & Tamura, 2009*; *Peeters et al., 2013a*). Table 1 mentions the reference sequences for new Rip genes. Two new hypothetical Rips were also identified; named Hyp17 and Hyp18 with two associated reference sequences (See Table 1). Considering that in the previous database (*Peeters et al., 2013a*), some Rips were only represented by pseudogenes, we corrected this by attributing new reference sequences to RipBA (strain Rs-10-244, sequence RS244_c002320), RipBE (strain YC40-M, sequence YC40-M_00170) and RipP3 (strain Rs-10-244, sequence RS244_c031810).

## Improved Rip-scanning pipeline

Thanks to the increased dataset of 102 total Rips (and 16 hypothetical T3Es), on a total of 155 genomes (totaling 140 different strains), we were able to generate new effector profiles for the improved prediction of these Rips and candidate Rips in newly available *Ralstonia* genomes. The "scan your genome" tool is available on the database website.

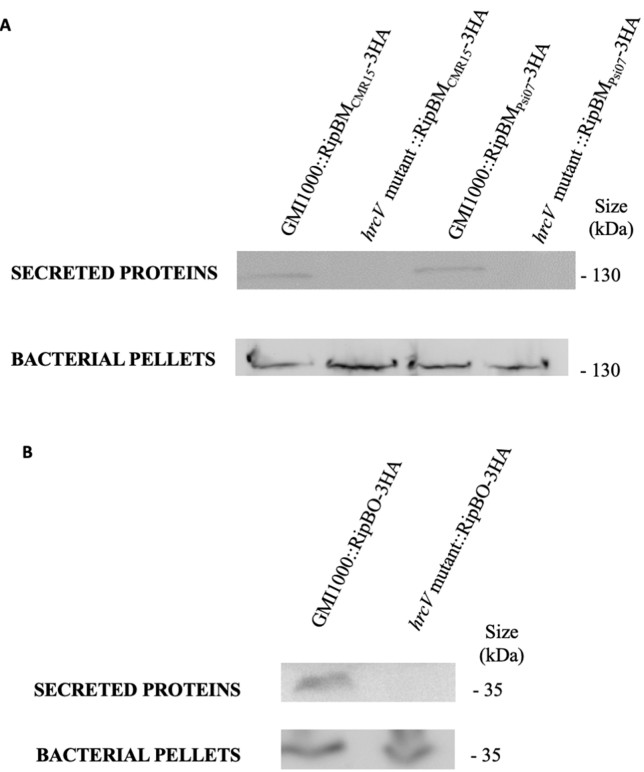

**Figure 2 RipBM and RipBO are secreted through the T3SS.** The wild-type strain and the *hrcV* mutant were transformed to express a RipBM<sub>CMR15</sub>-3HA, RipBM<sub>Psi07</sub>-3HA (A) or a RipBO-3HA (B) fusion protein. Secretion assays were performed and total proteins from bacterial pellets and proteins in the supernatants were detected by Western-Blot. Uncropped western-blot are displayed on Fig. S2.

For large dataset analysis, we prefer to be contacted directly (RalstoT3E-toulouse@inra.fr), to prevent server overload. A Blast tool, as well as all the files and results of predictions for the 155 genomes are also available on our website. A convenient tool is the availability of multifasta files for the nucleotide or protein sequences for a given Rip, containing the genome/strain sequences that one queried for comparison in the first place.

## Core effectors

We wanted to have a new look at the number of conserved Rips among this new diverse set of strains. As a principle, as more strains are compared, the smaller the core set of Rips will become. In order to have a pertinent set of strains to compare, we decided to limit the core comparisons to the "stringency 2" set of strains (the 84 distinct strains having less than 10 pseudogenes or frameshifted genes and, at the same time, more than 50 predicted Rip genes). Figure 3 shows a phylogenetic tree built using the *mutS* gene sequence of these 84 strains to be able to judge the relatedness of this set of strains. We are aware of the risk of excluding some strains based on this stringent selection. This could in particular be the case for the known strains that have seen genome reduction and hence have fewer T3Es. This is the case for the Moko disease, or blood disease bacterium BDBR229 (*Remenant et al., 2011*), and the *R. syzygii* clove-tree infecting and insect-transmitted
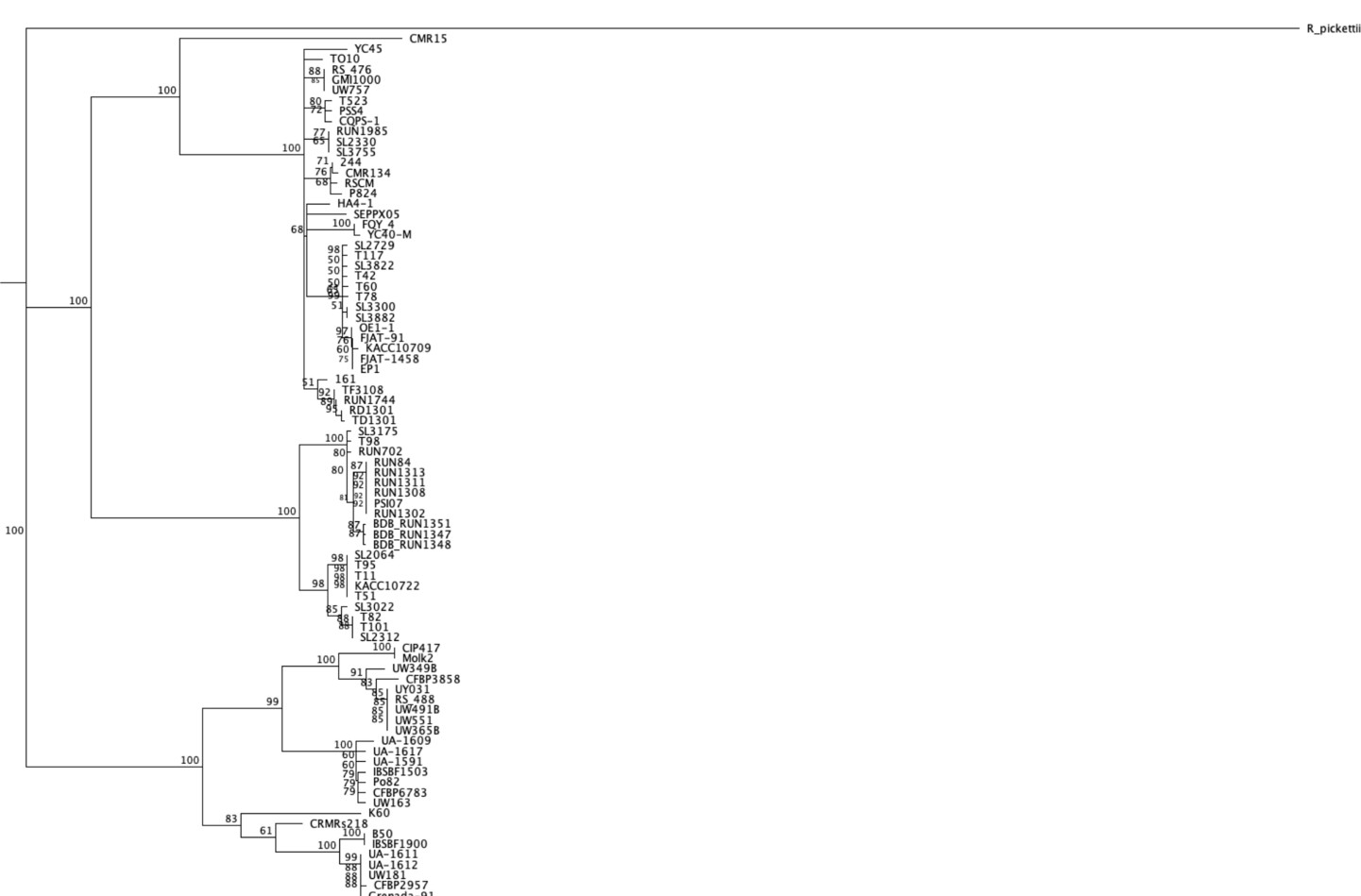

**Figure 3** ***mutS* alignment and phylogenetic tree on the set of 84 different strains.** A neighbor-joining tree was built using the *mutS* from *Ralstonia pickettii* as an outgroup. Bootstrap were performed on 100 replicates, only support higher than 50% displayed in the consensus tree.

R24 strain (*Remenant et al., 2011*), which each have respectively 54 and 48 Rip genes (as defined under the "stringency 2" criteria). Moreover, both strains are already left out under "stringency 1" criteria, for having more that 10 (respectively 20 and 12), frameshifted and pseudogenes.

We then decided to use the host of isolation as an interesting criterion to compare strains. Of course there are numerous examples of strains isolated on one host and later shown in laboratory settings to be able to infect other host plants. GMI1000, isolated from tomato (*Salanoubat et al., 2002*), was shown to be very well capable of wilting *Medicago truncatula* (*Vailleau et al., 2007*) or *Arabidopsis thaliana* (*Deslandes et al., 1998*). As laboratory settings are hard to compare between labs, and as thorough host-compatibility has been done only for a handful of strains, we preferred to stick with the host of isolation information, without excluding that the host range might be much wider for some strains, and restricted for others. We decided to compare the conservation of Rip repertoires among the 84 "stringency 2" strains, classifying them into hosts of

isolation; *Solanaceae* strains ("SOL"), tomato strains ("TOM" 15 strains), Eggplant ("EGG" nine strains), potato ("POT" 30 strains) and banana ("BAN" 15 strains). The larger, encompassing category being the Solanaceae group, with 58 strains (for the list of strains see Table S1). Table 2 indicates the list, per host-of-isolation category of the core set of Rip genes. For a set of *n* total strains, we decided to still consider core, the Rip genes present in the interval (*n*; *n*-5%) number of strains. For instance, for the 58 "SOL" strains, the core Rip genes are the ones present in 58 to 55 strains. Obviously, the larger the number of strains, the lower the number of conserved Rips. For instance, for the nine "EGG" strains, there are 44 strictly conserved Rips (in all nine strains), whereas there are only 27 conserved Rips in the 15 "BAN" strains (in 14 to 15 strains). Figure 4 show two Venn diagram comparing these sets of conserved Rip between host-of-isolation groups.

The set of 140 strains is evenly spread over the three newly defined *Ralstonia* species (see Fig. S1 for a *mutS* phylogeny (*Wicker et al., 2012*) of the 140 strain/155 genome sequences of this study). One possible caveat is the small number of phylotype III strains (only two strains: CMR15 and CFBP3059), now classified with phylotype I strains among the *R. pseudosolanacearum*. One interesting way to look at the conservation of Rips is to make specific species groups. The 84 "stringency 2" strains are well spread over the three phylogenetic groups: 38 are *R. pseudosolanacearum* strains (phylotypes I and III), 25 are *R. solanacearum* strains (phylotypes IIA and IIB), and 21 are *R. syzygii* strains (phylotype IV), see Fig. 3 for the *mutS* phylogeny of these 84 strains (*Wicker et al., 2012*). Table 3 represents the Rip distribution among these three species, together with the conservation in the total set of 84 strains. Figure 5 displays the Venn diagram corresponding to this triple comparison.

## DISCUSSION

In this work, we significantly updated the *Ralstonia* type III secretion effector database (*Peeters et al., 2013a*). This latter work, providing a new nomenclature for these essential virulence proteins was widely accepted and cited by the community. Here, we reported on the curation of new plant pathogenic *Ralstonia* strains, adding new T3Es to this database. These are represented by the new series from RipBJ to RipBQ, among which both RipBM and RipBO were shown in this work to be indeed secreted by the GMI1000 (*R. pseudosolanacearum*) T3SS. One of these newly defined Rips, RipBN, was identified for being an ortholog of the *Pseudomonas syringae* AvrRpt2 T3E (*Eschen-Lippold et al., 2016*), and recently shown to function similarly in triggering resistance in Ptr1-tomato lines (*Mazo-Molina et al., 2019*).

The newly defined Rip profiles (102 Rips and 16 Hypothetical Rips) were then used to predict the T3E repertoire of the 155 genome sequences available, representing a total of 140 different strains, compared to the 12 genomes previously available. This large set of strains allows us to provide an updated database with a better representation of each of the three phylogenetic clades of this species complex. These are: phylotypes I and III, or the newly proposed species named *R. pseudosolanacearum* (*Safni et al., 2014*), phylotypes IIA and IIB, or *R. solanacearum* and phylotype IV, or *R. syzygii*. For a better

**Table 2 List of Type III effectors conserved according to the host of isolation.**

| # strains | Eggplant 9 | Tomato 14–15 | Banana 14–15 | Potato 28–30 | Solanaceae 55–58 |
|---|---|---|---|---|---|
| RipA2 | 9 | 15 | | 30 | 58 |
| RipA3 | | 15 | | 28 | 55 |
| RipA4 | | 14 | | | |
| RipA5 | | 14 | 14 | 30 | 55 |
| RipB | 9 | 15 | 15 | 30 | 57 |
| RipC1 | | 14 | 15 | | |
| RipD | | | 15 | | |
| RipE1 | 9 | 14 | 15 | | |
| RipE2 | | | 15 | | |
| RipF1 | | 15 | 14 | 29 | |
| RipG2 | 9 | 14 | | | |
| RipG3 | | | 14 | | |
| RipG4 | 9 | | | | |
| RipG5 | 9 | 15 | 14 | 30 | 58 |
| RipG6 | 9 | 14 | 15 | 30 | 57 |
| RipG7 | | 15 | | 28 | 55 |
| RipH1 | 9 | 14 | 14 | | |
| RipH2 | 9 | 15 | 15 | 30 | 58 |
| RipH3 | 9 | 15 | | 28 | 56 |
| RipI | | 14 | 15 | | |
| RipJ | 9 | | | | |
| RipL | 9 | | | | |
| RipM | 9 | 15 | | 28 | 55 |
| RipN | 9 | 15 | | | 55 |
| RipO1 | 9 | | | | |
| RipQ | 9 | | | | |
| RipR | 9 | 15 | | 30 | 58 |
| RipS1 | 9 | | | | |
| RipS2 | | 15 | | | |
| RipS3 | | 15 | | | |
| RipS4 | | 14 | | | |
| RipS5 | 9 | 14 | 14 | 29 | 55 |
| RipS6 | 9 | | | | |
| RipU | 9 | 15 | | 30 | 58 |
| RipV1 | 9 | 15 | 15 | 28 | 56 |
| RipW | 9 | 15 | 15 | 29 | 57 |
| RipX | 9 | 15 | | 29 | 57 |
| RipY | 9 | 14 | | | |
| RipZ | 9 | 15 | | 29 | 57 |
| RipAA | 9 | 14 | | 29 | 55 |
| RipAB | 9 | 15 | 14 | 30 | 58 |

(Continued)

| # strains | Eggplant 9 | Tomato 14–15 | Banana 14–15 | Potato 28–30 | Solanaceae 55–58 |
|---|---|---|---|---|---|
| RipAC | 9 | 15 | 15 | | |
| RipAD | | 14 | | 29 | |
| RipAE | 9 | 15 | | | |
| RipAF1 | 9 | | | | |
| RipAI | 9 | 15 | 15 | 30 | 58 |
| RipAJ | 9 | 15 | 15 | 30 | 58 |
| RipAL | | | | 30 | |
| RipAM | 9 | 15 | | 30 | 58 |
| RipAN | | 15 | 15 | 30 | 57 |
| RipAO | 9 | 15 | 15 | 29 | 57 |
| RipAP | 9 | | | | |
| RipAQ | 9 | 15 | | 28 | 56 |
| RipAR | | 14 | 14 | | |
| RipAS | 9 | | | | |
| RipAT | | | 15 | | |
| RipAU | | | 14 | | |
| RipAV | 9 | | | | |
| RipAW | | 14 | | | |
| RipAX1 | | | 14 | | |
| RipAY | | 14 | 15 | 29 | |
| RipAZ1 | 9 | | | | |
| RipBA | 9 | | | | |
| RipBM | 9 | | | | |
| RipTAL | 9 | | | | |
| RipTPS | 9 | 14 | | 29 | 55 |
| Total CoreT3Es | 44 | 44 | 27 | 30 | 27 |

**Note:**
The # strains, indicate the total number of strains analyzed and the 5% tolerance. The numbers in each cell indicate how many strains actually harbor the cognate effector. Gray scale according to conservation between columns.

view of strain relatedness, *mutS* phylogenies are displayed in Fig. 3 (set of 84 strains), and in Fig. S1 (all 140 strains). In order to understand the contribution of these T3Es to the virulence of these bacteria on their host plant, it is particularly interesting to analyze which T3Es are conserved among the different strains. Our comparison results highlight two ways to explore these repertoires: either by host plant or by phylogenetic relatedness.

Ideally, each of the deeply-sequenced and well-annotated strains (the "stringency 2" list of 84 strains) should be tested on a panel of host plants in order to define their actual host range. As these data are not available we focused on the host of isolation as a limited but natural host definition factor. This is a strong limitation in this comparison, as it is reported or known (and shared through personal communications) that some strains are also compatible with other, and sometimes distantly related hosts. We decided to add the published information on the compatibility on other host plants in the Table S1. A few

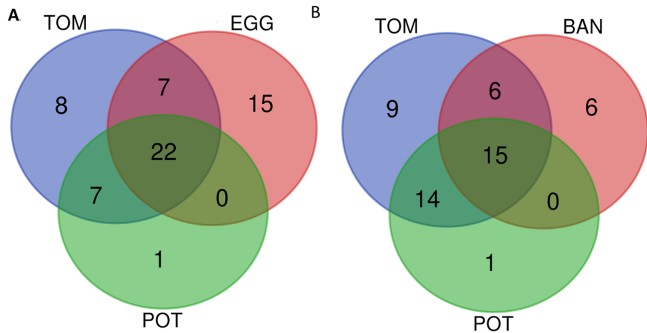

**Figure 4 Venn diagram of conserved T3Es among different sets of "host-of-isolation" defined strains.** (A) Comparisons of conserved T3Es among "TOM" (host of isolation: tomato), "EGG" (Host of isolation: eggplant) and "POT" (host of isolation: potato). (B) Comparison between "TOM" "POT" and "BAN" (host of isolation: banana). The lists of compared T3Es are visible in Table 2.

research groups added a significant amount of host-compatibility information for a set of strains (*Ailloud et al., 2015*; *Cho et al., 2018*; *Lebeau et al., 2011*). Other groups have performed host-compatibility experiments and shared this information with us, for example, tobacco strain CQPS-1 (*Liu et al., 2017*) is also mildly pathogenic on tomato (Prof. W Ding, 2019, personal communication), while the blueberry strains P816, P822 and P824 (*Bocsanczy, Espindola & Norman, 2019*), are very aggressive on tomato (Dr. DJ Norman, 2019, personal communication). In our repertoire comparison, we allow a tolerance of presence for the Rip in the interval of strains between the total number of strains *n*, and *n*-5%, this allows to compensate the effect of unequal set of strains to compare. Table 2 and Fig. 4A show that, unsurprisingly strains isolated from Eggplant "EGG" tomato "TOM" and potato "POT" share a significative number of their conserved T3Es (22 shared out of 44 "EGG" 44 "TOM" and 30 "POT"), this number is probably largely underestimated as we know that some of these *Solanaceae*-isolated strains are compatible with other *Solanaceae* (*Lebeau et al., 2011*). Another comparison shown (Fig. 4B) is between the "TOM" "POT" and banana "BAN" strains. Here, we can see that there could be more T3Es shared between "BAN" and "TOM" (21 out of 27 "BAN" strains) than between "BAN" and "POT" (15 out of 27 "BAN" strains). To evaluate this potential difference, one has to keep in mind that, although banana and *Solanaceae* are distantly related, it has been shown that nine out of the 27 "BAN" strains are also compatible with tomato and potato, when only one strain (BDBR229) was shown to be incompatible on these two *Solanaceae* hosts (*Ailloud et al., 2015*). When considering all *Solanaceae* (SOL) as host of isolation (58 strains from the "stringency 2" set of 84 strains), the core set of T3Es (as defined to be present in 55 to 58 strains) is represented by a list of 27 T3Es (see Table 2). It is only when host-compatibility is compared in detail with T3E repertoires that we can start to potentially associate the presence (or the presence of specific alleles) to be required (or deleterious) for specific host-compatibility (*Cho et al., 2019*; *Wang et al., 2016*).

A second and maybe less ambiguous way to compare lists of conserved T3Es is to group the strains by their phylogenetic origin. Table 3 summarizes the T3Es conserved within

DOI 10.7717/peerj.7346

**Table 3 List of Type III effectors conserved according to the phylogenetic origin.**

| # strains | Phylotype I and III 36–38 | Phylotype IIA and IIB 24–25 | Phylotype IV 20–21 | "stringency 2" 80–84 |
|---|---|---|---|---|
| RipA2 | 38 | (21) | 21 | 80 |
| RipA3 | 36 | | 21 | |
| RipA5 | | | 21 | |
| RipB | 37 | 25 | 21 | 83 |
| RipC1 | | 25 | | |
| RipD | | | 20 | |
| RipE1 | | 24 | | |
| RipE2 | | 24 | | |
| RipF1 | | 25 | 20 | |
| RipF2 | | 24 | | |
| RipG2 | 36 | | | |
| RipG4 | 38 | 24 | | |
| RipG5 | 38 | (23) | 21 | 82 |
| RipG6 | 37 | (23) | 21 | 81 |
| RipG7 | 37 | 24 | | |
| RipH1 | | | 20 | |
| RipH2 | 36 | 25 | 21 | 82 |
| RipH3 | 36 | | | |
| RipH4 | | | 20 | |
| RipI | | 25 | | |
| RipL | 38 | | | |
| RipM | | | 21 | |
| RipN | | | 20 | |
| RipO1 | | 24 | | |
| RipQ | 38 | | | |
| RipR | 38 | 24 | 20 | 82 |
| RipS2 | 36 | | | |
| RipS4 | 38 | | | |
| RipS5 | | | 21 | |
| RipS6 | 37 | | | |
| RipU | 38 | 25 | (18) | 81 |
| RipV1 | 38 | 25 | (19) | 82 |
| RipV2 | | | 20 | |
| RipW | 37 | 25 | 21 | 83 |
| RipX | | | 21 | |
| RipY | | | 20 | |
| RipZ | 38 | | 21 | |
| RipAB | 37 | 24 | 21 | 82 |
| RipAC | | 25 | | |
| RipAD | | 25 | | |
| RipAE | | 24 | | |
| Table 3 (continued). | | | | |
|---|---|---|---|---|
| # strains | Phylotype I and III 36–38 | Phylotype IIA and IIB 24–25 | Phylotype IV 20–21 | "stringency 2" 80–84 |
| RipAF1 | 36 | | | |
| RipAI | 36 | 25 | 21 | 82 |
| RipAJ | 38 | 25 | 21 | 84 |
| RipAK | 36 | | | |
| RipAM | 38 | (23) | 21 | 82 |
| RipAN | (35) | 25 | 21 | 81 |
| RipAO | 36 | 25 | 21 | 82 |
| RipAP | 36 | 25 | | |
| RipAQ | | | 21 | |
| RipAS | 36 | | | |
| RipAU | | | 20 | |
| RipAV | 36 | | | |
| RipAY | (35) | 25 | 20 | 80 |
| RipAZ1 | 36 | | 20 | |
| RipBF | | | 20 | |
| T3E_Hyp1 | | | 20 | |
| Total Core T3Es | 31 | 25 | 32 | 16 |

Note:
The # strains, indicate the total number of strains analyzed and the 5% tolerance. The numbers in each cell indicate how many strains actually harbor the cognate effector. Gray scale according to conservation between columns.

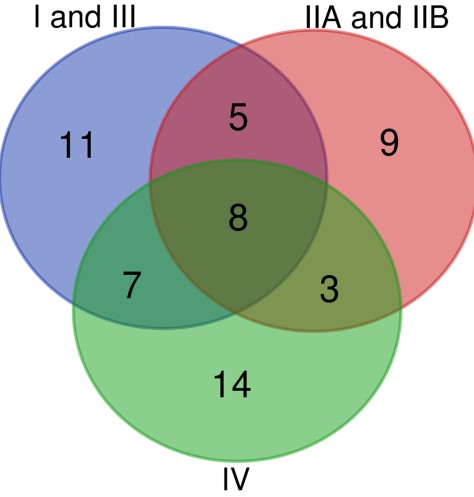

**Figure 5 Venn diagram of conserved T3Es among the different phylogenetic clades of strains.** Comparison of conserved T3Es between *R. pseudosolanacearum* (phylotypes I and III), *R. solanacearum* (phylotypes IIA and IIB), and *R. syzygii* (Phylotype IV) strains. The lists of compared T3Es are visible in Table 3.

each of the three phylogenetic groups of strains (*Wicker et al., 2012*). These groups are: the 38 strains from phylotypes I and III, or *R. pseudosolanacearum*; the 25 strains from phylotypes IIA and IIB, or *R. solanacearum*; and the 21 strains from phylotype IV or

*R. syzygii*; Fig. 3 displays the *mutS* phylogeny of these 84 strains. Some strong phylogenetic associated presence/absence can be highlighted, like the systematic presence in *R. pseudosolanacearum* and *R. syzygii* and systematic absence in *R. solanacearum* of the conserved T3Es RipA2, RipG5 and RipZ. Some T3Es are systematically associated with only one of these phylogenetic groups, like RipC1, RipI, RipAC, RipD with *R. solanacearum*; RipL, RipQ, RipS4 with *R. pseudosolanacearum* and RipA5, RipM, RipS5, RipAQ with *R syzygii*. A total of 16 T3Es are conserved among the phylogenetic groups (Fig. 5; Table 3). Eight of them are conserved in the different species: RipB (absent only in *R. pseudosolanacearum* CQPS-1 (*Liu et al., 2017*)); RipH2 (absent only in *R. pseudosolanacearum* RSCM isolated from *Cucurbita maxima* in China (*She et al., 2017*)); RipR (absent only in *R. solanacearum* UW181, a plantain banana strain (*Wicker et al., 2012*), and *R. syzygii* BDB_RUN1347, no host of isolation reported); RipW (absent only in *R. pseudosolanacearum* strain SL3822 isolated form potato in Korea (*Cho et al., 2018*); RipAB (absent only in *R. pseudosolanacearum* strain YC40-M, no host of isolation reported, and *R. solanacearum* strain MolK2 (*Remenant et al., 2010*)); RipAI (absent only in *R. pseudosolanacearum* strain HA4-1 a Chinese peanut strain); RipAO (absent only in *R. pseudosolanacearum* strain SL3755 isolated form potato in Korea (*Cho et al., 2018*)). The only strictly conserved T3E among these 84 strains is RipAJ. Eight others are slightly under-represented in one species out of the three (number of strains in which the T3E is present is indicated in brackets in Table 3). Among these two (2) are less conserved in *R. pseudosolanacearum* (Phylotype I and III): RipAN and RipAY; four (4) are less conserved in *R. solanacearum* (Phylotype II): RipA2, RipG5, RipG6, RipAM; two (2) are less conserved in *R. syzygii* (Phylotype IV): RipU and RipV1.

## CONCLUSIONS

This work describes the methods and strains used to build a comprehensive database of the type III effectors (T3Es) from the *R. solanacearum* Species Complex (*R. solanacearum*, *R. pseudosolanacearum* and *R. syzygii*). Representing a resource to both study and identify new allelic versions of specific T3Es, the database contains all the specific T3E sequences (102 T3Es and 16 hypothetical T3Es in over 155 strains), but also allows to identify new T3E orthologs by scanning DNA sequences (partial, shotgun or complete genomes) from original isolates.

## ACKNOWLEDGEMENTS

We wish to thank two anonymous reviewers and David Baltrus for helping use improve this manuscript with their thoughtful comments and suggestions.

### Funding

Cyrus Raja Rubenstein Sabbagh was funded by grants from Lebanon (the municipality of Nabatieh and the Association of Specialization and Scientific Orientation).

Alberto P. Macho was supported by the Shanghai Center for Plant Stress Biology (Chinese Academy of Sciences) and the Chinese 1000 Talents Program. We also received funding from the Laboratoire d'Excellence (LABEX) TULIP (ANR-10-LABX-41). The funders had no role in study design, data collection and analysis, decision to publish, or preparation of the manuscript.

## Grant Disclosures

The following grant information was disclosed by the authors:
Lebanon: the municipality of Nabatieh and the Association of Specialization and Scientific Orientation.
Shanghai Center for Plant Stress Biology (Chinese Academy of Sciences) and the Chinese 1000 Talents Program.
Laboratoire d'Excellence (LABEX) TULIP (ANR-10-LABX-41).

## Competing Interests

The authors declare that they have no competing interests.

## Author Contributions

- Cyrus Raja Rubenstein Sabbagh conceived and designed the experiments, performed the experiments, analyzed the data, prepared figures and/or tables, authored or reviewed drafts of the paper, approved the final draft.
- Sebastien Carrere conceived and designed the experiments, performed the experiments, analyzed the data, prepared figures and/or tables, authored or reviewed drafts of the paper, approved the final draft.
- Fabien Lonjon conceived and designed the experiments, performed the experiments, analyzed the data, prepared figures and/or tables, authored or reviewed drafts of the paper, approved the final draft.
- Fabienne Vailleau conceived and designed the experiments, analyzed the data, authored or reviewed drafts of the paper, approved the final draft.
- Alberto P. Macho conceived and designed the experiments, analyzed the data, contributed reagents/materials/analysis tools, authored or reviewed drafts of the paper, approved the final draft.
- Stephane Genin conceived and designed the experiments, performed the experiments, analyzed the data, prepared figures and/or tables, authored or reviewed drafts of the paper, approved the final draft.
- Nemo Peeters conceived and designed the experiments, performed the experiments, analyzed the data, prepared figures and/or tables, authored or reviewed drafts of the paper, approved the final draft.

## Data Availability

The genomes of the 155 strains are available in GenBank and references are available in Table S1, which also contains all the metadata necessary for re-running the analysis. All the

data can be downloaded using specific queries here: https://iant.toulouse.inra.fr/T3E.
Raw data files are also available (whole genome prediction files).

## Supplemental Information

Supplemental information for this article can be found online at http://dx.doi.org/10.7717/
peerj.7346#supplemental-information.

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
