# Peer review of "Pangenomic type III effector database of the plant pathogenic Ralstonia spp"

_PeerJ, doi:10.7717/peerj.7346_

## Round 0.1 · original submission · Minor Revisions

All reviewers are positive about the manuscript, but they have suggested minor corrections.

Reviewer 1 ·

Basic reporting

The language is clear and well written. The authors use of active voice is a nice change.

Experimental design

The rationale behind the analysis is clearly laid out. Treating frameshifts as fall calls seems questionable but the database makes the call clear and checking each frameshift is beyond the scope of this study, although systematic verification of frameshifts and pseudogene status really should be on the priority list.

The T3SS secretion assays are lacking a cytoplasmic leakage control.

Validity of the findings

The authors are clear as to why they have made their analysis choices and openly acknowledge any limitations.

Additional comments

This is a truly valuable and important resource for the community of MPMI researchers.

Reviewer 2 ·

Basic reporting

The authors improved a resource that is extremely valuable to the Ralstonia solanacearum community. I do not see any issues with this manuscript besides English language and grammar. In several instances, sentence structure, punctuation, singular/plural, and past/present tense need to be corrected. In some cases, these mistakes make it difficult to understand the meaning of sentences. I corrected some of these mistakes myself (see attached pdf) but I think it would be best if the authors still employed a professional editing service to do so in order to ensure a high quality manuscript.

Experimental design

No comment.

Validity of the findings

No comment.

Additional comments

Impressive work! I am looking forward to seeing this published.

Annotated reviews are not available for download in order to protect the identity of reviewers who chose to remain anonymous.

·

Basic reporting

no comment

Experimental design

no comment

Validity of the findings

no comment

Additional comments

In general, I found this manuscript to be well written and timely. The goal of this work is to update a previous iteration of the Ralstonia effector database and provide increased usability in terms of the functions and strains included. As such, it's a pretty straightforward descriptive work that *could* be useful to a variety of different research groups.

I really have no major qualms with the manuscript, the database and data are what they are and the authors do a good job of highlighting strengths and weaknesses in their approach.

The authors were careful in evaluating genomes for quality before inclusion, but I would suggest one more metric (since even complete genomes can be polluted with a bunch of frameshifts due to errors in long reads). Mick Watson came up with a handy way to at least quantify how often these frameshifts might be present in the genome (http://www.opiniomics.org/a-simple-test-for-uncorrected-insertions-and-deletions-indels-in-bacterial-genomes/) and I think it might be good to use a metric for the Ralstonia genomes. I'm guessing it won't change much, but something to think about as a platform for comparison of genome quality across sets.

Otherwise, some slight critiques:

L22: "causing the vascular" better as "causing vascular"

L52-53: "This bacterium is also particularly studied as it" could be rewritten a bit for clarity

L56: "bacterium in interaction" better as "bacterium during interaction"

L62: Legionella should be capitalized?

L78: "accommodating" better as "enabling"

L81: "This was performed taking into" better as "This analysis took into account:

L89: please define more clearly what you mean by "complete" genomes

L106-108: please include more details about why these genomes were of insufficient quality

L141: better as "and nine only contain"

L159-160: please rewrite this first sentence for clarity

L182: please include details of the PCR reaction conditions

L210 (and throughout): please evaluate all instances using the word "homology". Homology implies the exact same function. With effectors it's difficult, but for where you are just talking about sequence similarity, please replace the word "homology" with "similarity"

L269: ",the least Rips are conserved" better as "the lower the number of conserved Rips"

Please make sure that the words within the figures are legible (perhaps larger and a different font), I'm not sure that they are in this iteration.

---

## Round 0.2 · accepted · Accept

The manuscript was revised following reviewers' comments. I think it can now be accepted. However, new language errors were introduced. I spotted two, but there may be more. Search for the misspelled words critera and robosome. I recommend a thorough final language revision. Other than that, nice job!